# Benchtop Performance of Novel Mixed Ionic–Electronic Conductive Electrode Form Factors for Biopotential Recordings

**DOI:** 10.3390/s24103136

**Published:** 2024-05-15

**Authors:** Matthew Colachis, Bryan R. Schlink, Sam Colachis, Krenar Shqau, Brittani L. Huegen, Katherine Palmer, Amy Heintz

**Affiliations:** 1Battelle Memorial Institute, 505 King Ave., Columbus, OH 43201, USA; schlink@battelle.org (B.R.S.); shqauk@battelle.org (K.S.); palmerkm@battelle.org (K.P.); heintza@battelle.org (A.H.); 2UES, a BlueHalo Company, 4401 Dayton Xenia Road, Beavercreek, OH 45432, USA; brittani.huegen@bluehalo.com

**Keywords:** biopotential, mixed ionic–electronic conductive electrodes (MIEC), dry electrodes, EMG dry electrodes, bioelectrodes, non-invasive electrodes

## Abstract

*Background:* Traditional gel-based (wet) electrodes for biopotential recordings have several shortcomings that limit their practicality for real-world measurements. Dry electrodes may improve usability, but they often suffer from reduced signal quality. We sought to evaluate the biopotential recording properties of a novel mixed ionic–electronic conductive (MIEC) material for improved performance. *Methods:* We fabricated four MIEC electrode form factors and compared their signal recording properties to two control electrodes, which are electrodes commonly used for biopotential recordings (Ag-AgCl and stainless steel). We used an agar synthetic skin to characterize the impedance of each electrode form factor. An electrical phantom setup allowed us to compare the recording quality of simulated biopotentials with ground-truth sources. *Results:* All MIEC electrode form factors yielded impedances in a similar range to the control electrodes (all <80 kΩ at 100 Hz). Three of the four MIEC samples produced similar signal-to-noise ratios and interfacial charge transfers as the control electrodes. *Conclusions:* The MIEC electrodes demonstrated similar and, in some cases, better signal recording characteristics than current state-of-the-art electrodes. MIEC electrodes can also be fabricated into a myriad of form factors, underscoring the great potential this novel material has across a wide range of biopotential recording applications.

## 1. Introduction

The non-invasive measurement of human biopotentials via surface electrodes provides a simple yet informative method to capture the underlying electrophysiological processes in real-world environments. In recent years, biopotential measurements via wearable devices have become commonplace in a wide range of applications, including sports, remote health monitoring, and human–machine interface applications [1,2,3,4,5]. The appropriate selection of electrode type for non-invasive biopotential recording significantly impacts the quality and reliability of the recorded signals. Furthermore, different biopotential recording applications require different electrode form factors to enhance user experience.

Wet electrodes (those that utilize conductive gels to establish direct electrical contact with the skin) are traditionally used due to their ability to record high-fidelity electrophysiological data [6,7]. However, the gels applied to wet electrodes can irritate the skin if not applied properly. In specific instances, the gels can be difficult to remove after use. Electroencephalography (EEG) electrodes leave significant residue in the user’s hair, and electromyography (EMG) electrodes may require skin abrasion to fully remove the electrolytic gel. These gels can also dehydrate and coagulate with time, and the chloride layer on the electrode may deplete, leading to amplified signal detection interference and a decline in signal precision that necessitates replacement of the electrodes for continued measurement [8,9]. Unfortunately, many wet electrodes are not reusable, adding considerable cost to the end user. Together, these factors ultimately limit the usability and practicality of wet electrodes for biopotential recordings.

In contrast, dry electrodes are designed to eliminate the need for conductive gels or pastes, and they often use specialized materials or coatings to establish good electrical contact with the skin [10,11,12,13,14]. Eradicating the requirement to apply a conductive gel, dry electrodes offer the advantage of a faster setup with reduced skin preparation time, making them suitable for applications where convenience is crucial [10] and offering user comfort and ease of application. Unfortunately, most dry electrodes obtain slightly higher impedance (Z’) for signal recording compared to other electrodes [8,10,15]. Dry electrodes also exhibit higher electrode–skin impedance compared to wet electrodes, resulting in a reduced signal-to-noise ratio (SNR) and compromised signal fidelity [16].

As the demand for wearables and mobile electrophysiological applications has increased in recent years, so has the development of various dry electrode form factors. Textile electrodes offer improved user comfort and usability across a wide range of applications, but their durability and signal quality are often suboptimal for extended use [17]. Microneedle electrodes penetrate the upper layers of the skin, which greatly reduces skin impedance and improves signal quality [18]. These electrodes show promise for controlled biomedical applications, such as EKG, but a reliable fabrication method and protocol for precise measurement are lacking [19]. Finally, researchers have developed 3D-printed versions of Ag/AgCl electrodes for biopotential recording, such as EEG [20]. While these electrodes provide a unique solution to improve user comfort, they demonstrate reduced EEG signal quality compared to traditional wet electrodes, underscoring the need for novel electrodes that provide usability for a wide range of biopotential recording applications while also providing high-fidelity signal recording capabilities.

To address the challenges associated with current state-of-the-art dry electrodes for biopotential recording, we developed a novel and conformable dry electrode. It uses ionic and electronic conductors intertwined in an elastomeric matrix, which we refer to as a mixed ionic–electronic conductive (MIEC) electrode. It is readily fabricated through additive manufacturing and processed into various form factors specified to the desired applications [21]. Our previous work used sodium hyaluronate, exemplified by hyaluronic acid (HA), as an ionic conductor. This polymer has natural lubricating properties within the human body and has a high affinity for water retention. These properties of HA can aid in the interfacial resistance at the skin–electrode interfaces when combined with an electrical conductor like carbon nanotubes (CNTs) [22]. We used single-walled carbon nanotubes (SWNTs), as they have high bulk conductivity (exceeding 3000 S/cm in bulk films) and an elevated geometric aspect ratio (length-to-diameter ratio surpassing 105), a trait unparalleled among other carbon-based conductors [23,24,25,26,27]. This remarkable geometric aspect ratio allows MIEC electrodes to maintain their flexibility while simultaneously boasting exceptional bulk conductivity [22]. To modulate mechanical attributes and enhance skin interactions, the MIEC formulation incorporated acrylonitrile butadiene copolymer rubber (nitrile butadiene rubber, NBR). NBR, serving as the host matrix, bestows the electrodes with skin-like elastic compliance, a characteristic unattainable through HA and SWNTs alone. Since publishing our previous MIEC work [22], we have explored the integration of our MIEC material in different form factors relevant to common biopotential recording applications. The additional form factors allowed us to leverage different surface contacts of the electrode, whereas the only way to improve the surface contact of the previous MIEC iteration was through reduction of the modulus (i.e., adding more elastomer). The integration of this material may be incorporated into fabric, foams, and threads and can be drop casted, transfer printed, spayed, and screen printed on various textiles.

Material optimization was conducted in our previous work in terms of the MIEC material electrical conductivity [22]. The addition of a conductive additive (such as SWNTs) to the host material increased the electrical conductivity [22,28,29]. Increasing the loading of the filler might promote the formation of a percolating network of the conducting phase, resulting in a drastic increase in the conductivity. The percolation threshold is a function of the particle aspect ratio and dispersion in the matrix [30]. The electrical conductivity continues to increase as the loading increases, reaching a plateau that is a function of the filler resistivity and the junction [31].

The goal of this study was to compare the biopotential signal recording properties across different MIEC electrode form factors. We previously examined the potential of MIEC electrodes for non-invasive stimulation in electrotherapeutics [22]. Here, we aim to investigate how altering the morphology of the composite changes its recording capabilities. The electrodes were characterized through electrochemical impedance spectroscopy (EIS) to determine the interfacial charge transfer properties and EMG testing with an electrical phantom to analyze the SNR. The characterization of the signal properties of different MIEC electrode form factors may allow us to adapt and integrate MIEC electrodes for different applications across human performance, military, textile, and physiological monitoring while providing superior comfort, lower cost, reusability potential, and decreased weight compared to state-of-the-art wet electrodes.

## 2. Materials and Methods

### 2.1. Electrode Fabrication

Four different form factors of dry MIEC electrodes were fabricated to compare their signal recording properties (Table 1):MIEC casted on stainless steel (SS): MIEC-coated SS electrodes;MIEC foam: MIEC foam electrodes;MIEC sheet on silver (Ag) epoxy: MIEC elastomeric sheet connected with a flexible Ag epoxy backing;MIEC fabric: MIEC fabric electrodes that have the potential to be integrated into many textiles.

To compare the performance of these electrodes to the current state-of-the-art electrodes, we also analyzed SS dry electrodes (SS control) and Natus^®^ (Middleton, WI, USA) Ag-AgCl wet electrodes (Ag-AgCl) that were purchased.

#### 2.1.1. Fabrication of MIEC Slurry Masterbatch

A solution of HA sodium salt (MW = 15,000,000 to 18,000,000 Sigma-Aldrich^®^ (Burlington, MA, USA) from Streptococcus equi was prepared by mixing with deionized (DI) water (200:1 water: HA, by weight) and stirring with a stir bar for 24 h, resulting in a masterbatch of the HA solution with a concentration of 0.47%. The SWNT solution was received from OCsiAL (Leudelange, Luxembourg) (TUBALL^TM^ BATT H_2_O 0.4%) and further diluted with DI water (1:1 (*w*/*w*) ratio) to formulate a SWNT masterbatch. This solution was then combined with the HA masterbatch and Nipol^®^ LX370 (Zeon Chemicals L.P., Louisville, KY, USA) (37.5% total solids), acrylonitrile butadiene copolymer rubber (NBR), and placed in a 10 mL FlackTek SpeedMixer^®^ (Landrum, SC, USA) container. The ratios of SWNT-HA-NBR latex were kept consistent for all electrode form factors, resulting in SWNTs and HA loadings based on a total suspended solids (TSS) weight of 0.4 wt.% and 2.2 wt.%, respectively. A DAC 150 FVZ-K FlackTek SpeedMixer^®^ (Landrum, SC, USA) was used to thoroughly mix the sample, ensuring homogeneity. This final mixed solution was classified as the MIEC slurry masterbatch.

#### 2.1.2. Fabrication of Electrode Form Factors

*MIEC-Coated SS:* For the MIEC-coated SS electrode, 80 microliters of the MIEC slurry masterbatch was carefully drop casted onto a 12 mm diameter SS button. Prior to this deposition, the SS buttons underwent surface modification through grinding of their top surface using 240 or 280 grit sandpaper. This modified the material’s surface properties, allowing for better adhesion of the MIEC slurry. The SS buttons were then wiped with isopropanol and dried by patting the surface with Kimwipes^®^ (Kinberly-Clark Corporation, Irving, TX, USA). The solution was carefully dispensed on the button at a controlled pace to prevent bubble formation. Lastly, the samples were cured at room temperature for 24 h.

*MIEC foam:* MIEC foam electrodes were constructed by soaking and impregnating circular cut (d = 12 mm), super cushioning, ultra-comfortable polyurethane (PU) foam (McMaster-Carr^®^, Elmhurst, IL, USA) with the MIEC slurry masterbatch. This was completed by pouring the MIEC slurry masterbatch solution in a plastic bag, then placing the foam electrodes in the bag and letting the sample soak for 24 h at room temperature. The samples were then removed and set on a benchtop to cure at room temperature for 24 h. A two-part flexible electrically conductive pure Ag epoxy adhesive (Ag epoxy; 1:1 mix ratio), purchased from ConductiveX, was used to anchor an electrical wire to the backside of the MIEC foam electrode and let cure for 24 h at room temperature.

*MIEC sheet on Ag epoxy:* An MIEC sheet on Ag epoxy was created through casting the MIEC masterbatch slurry in a Delrin mold and letting it cure for 72 h. Samples were cut from the cured sheet to a 12 mm diameter, and Ag epoxy was deposited on the backside of the cut MIEC sheet to anchor the electrical wire to the electrode. This electrode was stretchable, flexible, and conformable.

*MIEC fabric:* MIEC fabric electrodes were fabricated by drop casting 300 microliters of the MIEC slurry master batch onto thin 12 mm copper mesh sewn into elastane fabric. The samples were then cured for 4 h. This process was repeated four times per electrode. A wire was sewn into the copper mesh prior to deposition, which allowed for a solid connection from the soft to hard interface. The wire was impregnated into the cured solution for proper electrical connection to the electrode. This electrode was flexible, conformable, and durable.

In addition to these four novel MIEC electrode form factors, we also evaluated SS electrodes (SS control) and Natus Ag-AgCl electrodes (Ag-AgCl) for comparison (Table 1). We reported the cytotoxicity of the MIEC formulation in our previous study [22].

#### 2.1.3. Synthetic Skin Fabrication

Synthetic skin plates with wires embedded into them (for artificial EMG measurements) [32] were prepared with a solution of 4.5% *w*/*v* agar and 0.97% *w*/*v* NaCl, prepared in DI water, which was the same formulation used by Lam et al. [32,33]. This solution was then heated via a 20 min sterilization time liquid autoclave cycle to dissolve the agar powder. After cooling slightly (approx. 80 °C), 18 mL of the solution was aliquoted into 100 mm × 15 mm petri dishes and allowed to cool for several seconds. Two electrical wires were placed on top of the agar layer, with exposed ends in the center of the plate. An additional 22 mL of the solution was added on top of these wires. The wires were taped into place on the side of the petri dish to ensure they did not move while cooling or during use. All plates were allowed to fully cool until reaching room temperature, then stored inverted and sealed with parafilm at 4 °C until use. This conductive medium was used for EIS and artificial EMG.

### 2.2. Electrode–Skin Interfacial Charge-Transfer Characterization via Electrical Impedance Spectroscopy (EIS)

To characterize the impedance of each set of experimental electrodes, we performed an electrode–skin interfacial charge-transfer characterization through EIS, which involves applying small alternating current (AC) signals across the electrodes and measuring the resulting voltage responses over a wide frequency range. Here, we used EIS to determine an impedance vs. frequency sweep of the electrodes. An impedance sweep at frequencies of 655.35 kHz–1 Hz was performed three times on each set of electrode form factors. The interfacial charge transfer of the electrodes was also investigated to determine whether there was any correlation to the electrodes’ SNR.

For a conductive medium, synthetic skin was utilized. A pair of wires was embedded within the synthetic skin for the transmission of simulated data to be recorded by the electrode samples (refer to Section 2.4 for full details). For each set of form factors (including SS and Ag-AgCl), the two electrodes were placed on the artificial skin for testing and spaced 25.4 mm apart, which is a common electrode distance to use [34,35]. The electrodes were placed firmly on the skin and a plastic dish, with a constant pressure of 4.7 mmHg on top of the electrodes to ensure consistent electrode–skin contact. Each electrode was connected to the Potentiostat Galvanostat device and the Impendence Analyzer using alligator clips. The setup prior to testing had a direct current (DC) potential of 0 V, an AC amplitude of 5 mV for polarization, an initial frequency of 655.35 kHz, and a final frequency of 1 Hz. The resulting impedance vs. frequency was plotted, the interfacial charge-transfer values were fitted using a Randell’s circuit (Figure 1a), and the fitted estimated values were analyzed using the ZPlot^®^ (Scribner, LLC, Southern Pines, NC, USA) fitting program. A Randell’s Cell model consists of a charge-transfer resistor (Rd) in parallel with a double-layer capacitor (Cdl) and a solution or bulk resistance (Rs) in series, where Rs often represents the skin’s bulk resistance or solution resistance, Rd represents the electrode interface resistance, and Cdl represents the interfacial capacitance of the electrode–skin interface [36].

### 2.3. Ground-Truth Biopotential Measurement Using an Electrical Phantom

To provide a repeatable comparison of the signal quality across the different electrode form factors, we used an electrical phantom setup with ground-truth sources of simulated EMG activity (Figure 1b). Though MIEC electrodes can record a range of human biopotentials, we selected EMG for this pilot data collection due to its robust signal amplitude compared to other physiological signals recorded from the skin, such as EEG or electrocardiogram (EKG). An electrical phantom consists of a conductive material with embedded wires used to transmit known signals to the phantom surface for recording [32,39]. Phantoms can be used for objective, ground-truth evaluations of various aspects related to signal quality, such as crosstalk and motion artifacts [32]. Here, we used the synthetic skin described in Section 2.1.3 to objectively evaluate the SNR with each electrode composition. A pair of wires embedded within the conductive gelatin acted as an antenna to broadcast the simulated EMG signals. The simulated EMG data were created using physiologically relevant parameters for human muscle and have been used previously to test EMG technology with a phantom device [32]. We used custom MATLAB (MathWorks, Natick, MA) scripts and a data acquisition module (National Instruments, Austin, TX) to transmit the simulated data through the synthetic skin.

Similar to the experimental setup for EIS, we placed each pair of electrodes 25.4 mm apart on the surface of the synthetic skin. Because we were transmitting simulated EMG data, we selected an interelectrode spacing typical and appropriate for surface EMG recording in a bipolar configuration [40,41,42]. We placed a third reference electrode 25.4 mm apart from the others, with the embedded wire ends centered between all three electrodes. To standardize the comparisons, we used an SS reference electrode for recordings with each of the form factors (Figure 1b). The hydrogel electrode (Ag-AgCl) was disregarded as the reference electrode due to its moisture absorption properties; therefore, the SS electrode was employed as the reference electrode to ensure a consistent result.

We broadcasted the 10 s simulated data file through the embedded wires and recorded the broadcast signals at the surface with each pair of electrodes. The data file consisted of seven pulse trains, where each pulse train consisted of ~0.5 s of signal and ~0.5 s of no signal. The signal was acquired using an RHD USB Interface Board (Intan Technologies, Los Angeles, CA, USA) and an RHD Electrophysiology Amplifier Chip (Intan Technologies, Los Angeles, CA, USA) configured for differential recording with a common reference. For each recording, we then filtered the recorded signals at a 60 Hz notch and a 10th order Butterworth bandpass at 130–400 Hz [43] and calculated the SNR. The SNR was calculated for each pulse train by dividing the root mean square (RMS) of the signal by the RMS of the noise, yielding seven SNR values for each electrode sample. The SNR values for each electrode sample were then normalized to average SNR for SS to account for day-to-day variability.

### 2.4. Mechanical Properties: Tensile Test

The mechanical properties of the MIEC sheets were evaluated using tensile testing under ASTM D412, as cited in the surgical glove standard [44] Dogbones of the MIEC sheet were cut out, and a basic tensile test with an extensometer was performed on the samples using either the Instron 5982 or 5564 (Instron, Norwood, MA, USA) models. A 1-inch gauge length was utilized with a strain rate of 2.00 inches per minute at room temperature. The corresponding data were then analyzed and extracted using the Bluehill Universal Version 4 (Instron, Norwood, MA, USA) software.

### 2.5. Scanning Electron Microscopy

The scanning electron microscopy (SEM) topology images of the MIEC film were acquired using an Apreo SEM (Thermo Fisher, Waltham, MA, USA) device with a 10 kV accelerating voltage and spot size of 4.0. Prior to imaging, 100 µL of the MIEC slurry was drop casted onto a glass slide and allowed to dry for 24 h before placing the glass slide sample onto an SEM stub. Post processing on the images was conducted using imageJ 1.53a (University of Wisconsin in Madison, WI, USA).

## 3. Results

### 3.1. Impedance Measurements

The Bode plot of the impedance vs. frequency sweep of the different electrodes was recorded through EIS (Figure 2a). Impedance values at 100 Hz, which is the working frequency for EMG [33,45], were then found from the impedance vs. frequency sweep (Figure 2b). It was observed that as the frequency decreases, the impedance increases.

Ag-AgCl had the lowest impedance (0.19 ± 0.01 kΩ), and the MIEC electrode on the SS electrode (0.29 ± 0.02 kΩ) had the second-lowest impedance at 100 Hz of all the electrodes and the lowest impedance of all the dry electrodes. All of the electrodes yielded impedances lower than 80 kΩ at 100 Hz, which is consistent with other COTS dry electrodes [11,46]. The MIEC fabric electrode demonstrated a low impedance at 100 Hz (1.90 ± 0.10 kΩ), falling in the range of a wet electrode performance. The MIEC foam electrode had the highest impedance at 100 Hz (68.53 ± 9.29 kΩ).

### 3.2. Artificial EMG Signaling SNR

To evaluate the signal quality across the electrode samples, we calculated and compared the SNR values based on artificial EMG signal recordings (Figure 3a). We observed that the MIEC electrode on SS yielded the highest normalized SNR (1.43 ± 0.09), surpassing both the Ag-AgCl (1.22 ± 0.05) and SS controls (1.00 ± 0.03). Interestingly, the MIEC fabric and MIEC electrode on Ag epoxy yielded normalized SNR values comparable to the SS control (1.00 ± 0.06, 0.96 ± 0.05, respectively). The MIEC foam yielded the lowest normalized SNR (0.13 ± 0.00). Upon further examination of the filtered artificial EMG signal (Figure 3b), the MIEC foam had the highest noise floor compared to the other electrode samples. However, the MIEC foam still had peak-to-peak voltages during signal bursts that were higher than most other samples. The Ag-AgCl and SS controls had the lowest noise floors, though still comparable to all of the MIEC formulations except the MIEC foam.

### 3.3. Interfacial Charge Transfer of Electrodes

Interfacial charge transfer was performed through EIS on the electrodes on the same surrogate skin used for artificial EMG testing. The Nyquist plot output was fitted by a Randell's Cell model, which consists of a charge-transfer resistor (Rd) in parallel with a double-layer capacitor (Cdl) and a solution or bulk resistance (Rs) in series [36]. The interfacial charge resistance measurements were performed through ZPlot^®^ 3.5i by modeling the circuit as a Randell’s Cell. Electrode interfacial charge values, impedance at 100 Hz, and normalized SNR were then reported (Table 2).

The analysis shows that the MIEC electrode on SS has the lowest interfacial impedance (64 ± 3 Ω), lower than both the SS (65 ± 1 Ω) and Ag-AgCl (115 ± 9 Ω). The highest interfacial charge resistance value present was seen in the MIEC foam electrode (41 ± 6 kΩ). All the other electrodes besides the foam electrode had an interfacial resistance value under 7 kΩ. The interfacial capacitance was seen to be the highest in the MIEC SS electrode (124 ± 2 nF), followed by the SS electrode (121 ± 5 nF). Besides the MIEC foam electrode, which had the lowest interfacial capacitance, all the electrodes had an interfacial capacitance value above 1 nF.

### 3.4. Mechanical Properties of MIEC Sheet

A stress vs. strain curve (Figure 4) was created through tensile tests on the MIEC sheet material. Corresponding mechanical strength values for the material were calculated (Table 3).

The average modulus, average ultimate tensile strength (UTS), and the average strain at break were 165.50 ± 21.70 MPa, 0.85 ± 0.01 MPa, and 567.51 ± 8.45%, respectively.

### 3.5. SEM Imaging

The SEM imaging of the MIEC sheet was analyzed at 2000× magnification and 10,000× magnification (Figure 5). The SWNT diameter was calculated using imageJ and was seen to be 61.4 ± 19.0 nm, with fiber lengths above 5 microns.

## 4. Discussion

This study explored the signal acquisition properties of different novel MIEC form factors for biopotential recordings. We analyzed and compared electrical material properties and signal quality across multiple electrode samples and identified MIEC form factors that demonstrate to be the most promising for dry electrode recording applications. These results provide critical insights for the use of MIEC electrodes in different form factors, optimized for specific applications.

### 4.1. Impedance and Interfacial Charge-Transfer Properties of Electrodes

The impedance of the electrode–skin interface was characterized through EIS by a complex value that included both resistance (real part) and reactance (imaginary part) [47]. Our previous work investigated changes to the composite formulation (e.g., relative ratios of ionic and electronic contribution) on impedance and charge transfer. In this study, we kept the formulation constant but changed the form factor, understanding that different form factors have different functional utilities. They also make different electrical contacts with the skin, with different conformabilities, and thus have the potential for forming good contact with the skin and have different surface areas and porosities. It is this balance in trades that we studied.

*Impedance:* We investigated the impedance of the electrodes at 100 Hz, as this is the peak frequency component for EMG recording [33,45]. It was seen that the interfacial impedance decreased as a function of increasing frequency. Lower impedance is an indication of good electrical contact and efficient signal transmission [48]. At 100 Hz, all MIEC electrode form factors, with the exception of the MIEC foam electrode, had exceptionally low impedances (<11 kΩ), which fell well below the impedance range of previously reported dry electrodes in the literature (17–169 kΩ at 100 Hz) [8,10,11,46]. The MIEC SS and fabric electrode had impedance values within the range of a wet electrode (<5 kΩ) [49,50]. We attribute the low impedances to their excellent conformability to the synthetic skin, higher double-layer capacitance, and reduced polarization resistance. The MIEC foam electrode had an impedance that was within the dry electrode range (of 69 ± 9 kΩ at 100 Hz) but was significantly higher than the other investigated electrodes. We hypothesized that the air gaps or voids within the foam structure, which act as insulators, substantially decrease the true electrical contact area [51]. An inadequate contact area between the electrode and the skin increases the impedance, resulting in reduced signal quality [46]. This electrode had the most potential for further optimization, as pressure to the foam electrodes on the synthetic skin should decrease the impedance and would be worth exploring in future work. The relatively low impedance results of our MIEC electrode form factors outperform those currently reported for dry electrodes [8,10,11,46] and had comparable values to that of the state-of-the-art wet Ag-AgCl electrodes [6,7,50].

*Interfacial charge transfer:* Electrode interfacial charge resistance or polarization resistance refers to the resistance that arises due to the polarization of the electrode when a voltage is applied [52,53,54]. It was seen that the electrodes with the highest interfacial capacitance values also had the lowest interfacial resistance and impedance values. This was consistent in all the electrodes tested; increasing the interfacial capacitance and decreasing the interfacial resistance of the electrode resulted in a low impedance. The electrode–skin equivalent capacitance might be influenced by the electrode’s adherence and conformance to the skin, as higher conformity leads to an increased contact area at the interface, resulting in our electrode’s improved interfacial resistance and capacitance performance [55].

### 4.2. Signal Quality—SNR (Limited to EMG Recording)

Apart from the MIEC foam electrodes, the different MIEC electrode form factors recorded simulated EMG data with similar SNRs compared to state-of-the-art electrodes (Ag-AgCl and SS). The poor SNR of the MIEC foam electrodes was illustrated in the visibly high baseline noise during recording with the electrical phantom setup (Figure 3b). As discussed, increased pressure on the foam electrodes may have reduced impedance and improved the SNR. However, the simulated bursts of EMG activity were still visible with a larger amplitude than the increased baseline noise. This suggests that, even in the presence of suboptimal pressure and an experimental setup, foam electrodes have potential for use in applications with biopotentials that have larger signal amplitudes (e.g., EMG). The performance of all electrodes could also be further improved by optimizing the electrode size based on the desired application. For example, fabric electrodes embedded within garments have an improved SNR when the electrode size is increased [56].

### 4.3. Relationship of Interfacial Charge Transfer to SNR

The electrode SNR was influenced by various factors, including the interfacial resistance and double-layer capacitance of the tested electrodes. The quality of the measured signal is impacted by both electrical characteristics [57]. At the interface, the double-layer capacitance is inversely related to the impedance, which can have a positive impact on the SNR through the facilitation of signal amplification and noise filtering [52]. Degradation on the SNR can occur at high interfacial resistance by limiting the ability of the electrode to accurately capture expeditious changes in the bioelectric signals, indicating that low Rd is desirable for biopotential applications [10,58]. Lower Rd and higher Cd values result in a lower total impedance based on a nonlinear fitting method applying the least squares approach, and it was used to estimate the different values of an electrical circuit model [55,59]. This result was confirmed by our results, with the best-performing electrode displaying these characteristics. Although the foam and Ag epoxy MIEC electrodes did not exceed the control electrode values, their flexibility and comfort present considerable benefits. Understanding these relationships will allow for better optimization of MIEC form factors for specific applications.

### 4.4. Mechanical Strength of MIEC Material

The mechanical strength values (Table 3) are comparable to the average elastic modulus of human skin (98.97 ± 97 MPa) [60,61] and outperform that of the state-of-the-art hydrogel Ag-AgCl electrodes (0.21 MPa) [62]. The large variations in human skin properties significantly depend on the orientation, location, age of person, anisotropic nature, sensitivity of the biological tissue being tested, and different mechanical testing parameters [60,63]. Although our material had a lower UTS (0.85 ± 0.01 MPa) than skin (27.2 ± 9.3 MPa) [60], it had a significantly larger average strain at break (567.51 ± 8.45%) compared to the average failure strain of biological skin (25.45 ± 5.07%) [60] and state-of-the-art hydrogel Ag-AgCl electrodes (105%) [62]. This allows the MIEC material to have superior flexibility and durability, potentially offering enhanced performance in applications that require extensive deformation or stretching.

### 4.5. SEM Analysis of MIEC Sheets

Based on the SEM images (Figure 5) obtained for the self-standing MIEC sheet electrode, the CNT conductive phase was uniformly distributed into the NBR matrix. This is not easily seen by the SEM images due to the low contrast in the electron beam. As observed, the 10k magnification SEM image (Figure 5b) displayed bundles with a diameter on the nanoscale (61.4 ± 19.0 nm) and fiber lengths above 5 microns. There was no directionality or evidence of alignment of the nanofibers in the images. In addition, the SWNTs were entangled, thus forming an interconnected network, ensuring relatively high electrical conductivity at very low solid loadings (below 0.5%) [22,30].

### 4.6. Other Potential Signal Acquisition Applications

To objectively compare the biopotential recording characteristics of each electrode form factor, we used simulated EMG data based on our team’s expertise and experience analyzing EMG signal properties. However, the design and composition of MIEC electrodes extends well beyond the measurement of muscle activity and provides the capability to capture a broad range of human physiological data, including but not limited to brain activity (via EEG) and electrical activity from the heart (via EKG). Future efforts are needed to test electrode performance for these applications.

Fabric electrodes hold significant potential in EMG and EKG applications, where garment integration is necessary for unobtrusive bio-signal monitoring during physical activity. While there are many promising e-textiles either on the market or in development [64,65,66,67,68,69], they are limited in the signal fidelity required for more complex physiological analyses. Specifically, many of these solutions yield a lower SNR, impacting fine motor decoding (for EMG applications) and/or cardiovascular health monitoring (for EKG applications).

However, other EMG applications require more robust hardware for data collection, making SS and coated Ag epoxy important form factors. For example, our team has developed a wearable garment that is populated with SS electrodes for use as a rehabilitative and/or assistive device for patients with neurological disorders [35,70,71]. While this system may benefit from fabric electrodes as an assistive device used by patients throughout daily activities, a more robust version with SS or Ag epoxy electrodes is important for use in a clinic to ensure high-fidelity signal recording. These types of systems may be further improved by developing advanced conduction enhancer coatings, such as MIEC coatings, to improve signal quality.

Foam electrodes hold significant potential in EEG applications that necessitate helmet-based brain signal monitoring. This approach mitigates the undesirable impact of wet hydrogel electrodes, which can negatively affect hair and compromise electrode–skin contact, and significantly reduces the need for electrode preparation.

### 4.7. Future Directions

The continual evaluation of these form factor properties will enable advanced systems for signal acquisition across many biopotential applications. In future studies, we plan to explore material properties more systematically for each MIEC form factor across a spectrum of MIEC formulations. This study was designed to demonstrate a proof of concept for various form factors, and as such, we chose to explore only a single formulation based on prior work. While this formulation demonstrated promise across multiple form factors, it was not systematically optimized for various physiological signal recording applications. Furthermore, this study was limited to artificial skin testing. Future testing in human subjects will be required to characterize and validate the real-world use of MIEC electrodes. It is also important to consider how the performance of MIEC form factors compares to previously published studies, as well as new electrode compositions developed for real-world applications.

### 4.8. Study Limitations

We acknowledge a few limitations in the design of this study. First, the interfacial charge-transfer properties of the experimental electrodes can be affected by several factors, such as electrode positioning, fluctuations in electrode mass, inadequate pressure application onto the electrode, variations in synthetic skin temperature, relative humidity, slight discrepancies in skin thickness among samples, and inherent distinctions in batch properties between the acquired SWNTs and HA from the supplier. We made every effort to ensure a consistent experimental setup across electrodes. Additional analyses in more controlled settings could help refine the selection of electrodes for different biopotential applications. We also could have more accurately modeled the double-layer capacitance by fitting the Randell’s Cell with a constant phase element (CPE) instead of a double-layer capacitor [72,73]. Finally, our experiments were performed in laboratory environments under controlled, stationary conditions. Real-world applications introduce a host of factors that can affect biopotential signal quality, such as motion artifacts, increased skin impedance, and stress. Future studies should investigate the effects of these factors on the signal performance of various MIEC form factors.

### 4.9. Concluding Remarks

Choosing the appropriate electrodes for biopotential measurements depends on the specific requirements of the application. Ag-AgCl electrodes remain a popular choice for their reliability and ease of use, and disposable electrodes are convenient for short-term use, while dry electrodes offer a promising solution for quick setup. The ideal electrodes are determined based on user preferences and signal quality, considering the following characteristics: (1) low electrode–skin contact impedance; (2) ease of use and convenience; and (3) the ability to produce high signal quality with minimal artifacts [14]. We have successfully created a novel dry MIEC electrode, constructed in various form factors, which outperforms the current state-of-the-art dry SS and wet Ag-AgCl electrodes’ normalized SNR for synthetic EMG. Creating a true dry electrode that has a similar SNR to the state-of-the-art dry SS electrode and wet Ag-AgCl electrode allows for an electrode with increased wearability and comfort through its conformability and flexibility, discrete profile, decreased weight, and ease of integration into textiles and wearables.

## Figures and Tables

**Figure 1 sensors-24-03136-f001:**
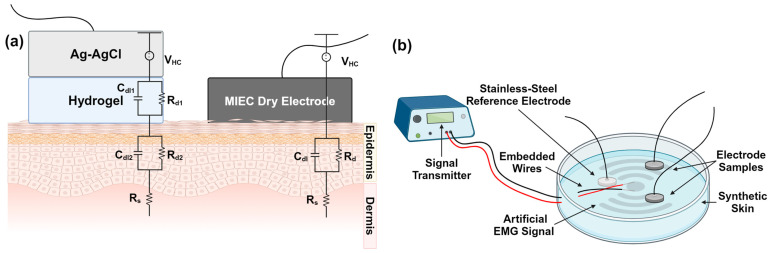
(**a**) Randell’s Cell model for the interfacial charge transfer parameters for wet and dry electrodes on human skin [37,38]. Randell’s Cell model consists of a charge-transfer resistor (Rd) in parallel with a double-layer capacitor (Cdl) and a solution or bulk resistance (Rs) in series, where Rs represents the skin’s bulk resistance, Rd represents the electrode interface resistance, Cdl represents the interfacial capacitance of the electrode–skin interface, and VHC is the half-cell potential [36,38]. (**b**) Electrical phantom setup with ground-truth sources of the simulated EMG activity. Created with BioRender.com (accessed on 27 March 2024).

**Figure 2 sensors-24-03136-f002:**
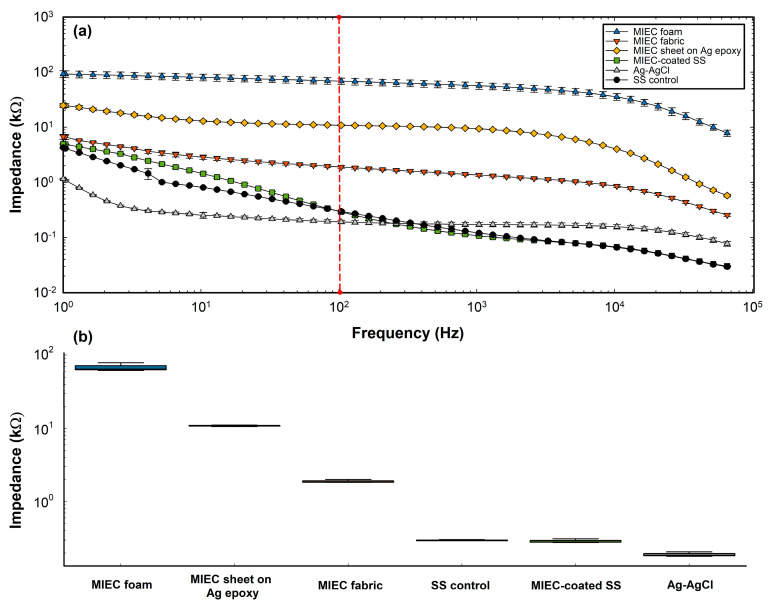
(**a**) Impedance versus frequency (Bode) plot for each electrode form factor on the artificial skin. Frequencies (x−axis) and impedance (y−axis) are shown on a logarithmic scale. Error bars represent the standard deviation of the three runs per electrode. (**b**) Impedance of tested electrodes at 100 Hz (created with Python 3.7 and SigmaPlot 10.0).

**Figure 3 sensors-24-03136-f003:**
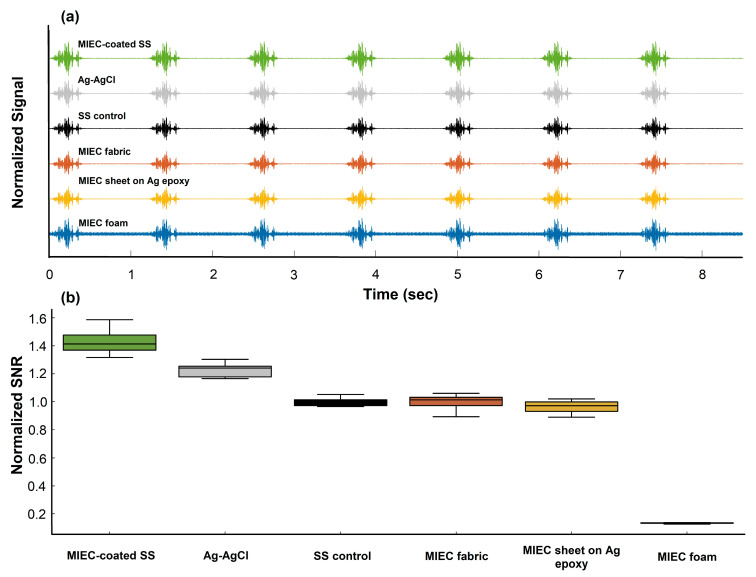
(**a**) Normalized SNR for tested electrodes. SNR values for each electrode sample were normalized to average SNR for SS control. Data presented as box-and-whisker plot. (**b**) Normalized Signal Plots. Signal was normalized to the min and max voltage values of SS control signal to account for day-to-day variability (created with Python 3.7 and SigmaPlot 10.0).

**Figure 4 sensors-24-03136-f004:**
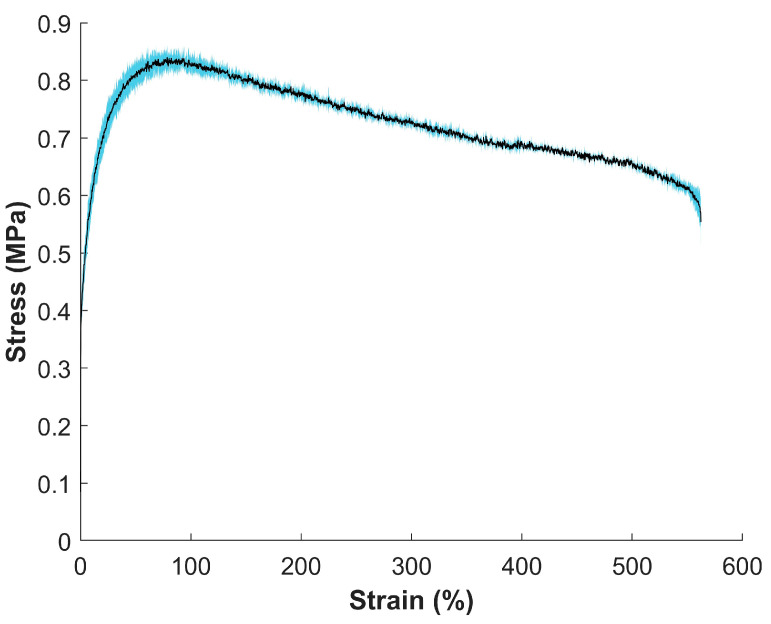
Stress vs. strain curve for the MIEC electrode material. The black line is the average of three different MIEC tensile tests with a strain rate of 2.00 in/min under room temperature conditions, and the blue shaded region is the standard deviation of the sample population. Corresponding mechanical strength values can be seen in Table 3.

**Figure 5 sensors-24-03136-f005:**
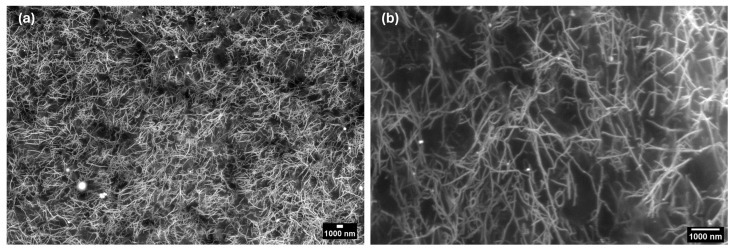
Scanning electron microscopy (SEM) images of the MIEC sheet electrode using 10 kV acceleration voltage at different magnifications: (**a**) 2000× (2 k) magnification and **(b**) 10,000× (10 k) magnification. Image analysis, color correction, and brightness/contrast were performed using imageJ 1.53a.

**Table 1 sensors-24-03136-t001:** Overview of MIEC form factors.

**MIEC-Coated SS** (MIEC slurry masterbatch drop casted on COTS SS button) Diameter: 12 mm Thickness: 0.48 ± 0.04 mm	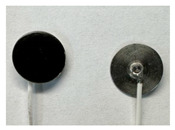	**MIEC Fabric** (MIEC slurry masterbatch drop cast and impregnated on elastane with copper mesh; flexible, conformable, durable) Diameter: 12 mm Thickness: 1.77 ± 0.22 mm	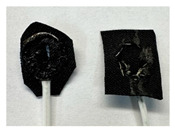
**MIEC Foam** (MIEC slurry impregnated in COTS PU foam) Diameter: 12 mm Thickness: 4.95 ± 0.47 mm	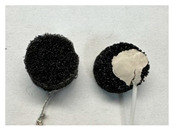	**SS Control** (Custom-made SS electrode) Diameter: 12 mm Thickness: 0.45 ± 0.02	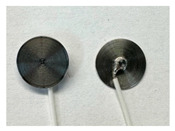
**MIEC Sheet on Ag Epoxy** (Ag epoxy cured on MIEC sheet; stretchable, flexible, conformable) Diameter: 12 mm Thickness: 1.43 ± 0.16 mm	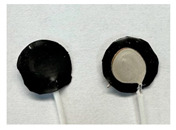	**Ag-AgCl** (Modified COTS Natus Ag-AgCl) Diameter: 12 mm Thickness: 1.24 ± 0.11 mm	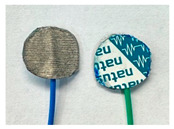

**Table 2 sensors-24-03136-t002:** Summary of electrode characteristics for each material.

Electrode	Skin Adherence	Liquid Phase–Gel Interface	R_s_ (Ω)	R_d_ (Ω)	C_dl_ (nF)	Z’ at 100 Hz (Ω) ^(a)^ *	Normalized SNR ^(a)^
**MIEC Foam**	Mechanical	No	3783.33 ± 110.57	41,496.00 ± 5820.52	0.18 ± 0.03	68,534.67 ± 9285.69 (6)	0.130 ± 0.004 (VI)
**MIEC Sheet on Ag Epoxy**	Mechanical	No	300.47 ± 20.88	6825.67 ± 191.94	1.78 ± 0.08	10,872.33 ± 263.12 (5)	0.960 ± 0.048 (V)
**MIEC Fabric**	Mechanical	No	158.57 ± 9.93	913.43 ± 58.07	7.96 ± 0.54	1899.97 ± 99.49 (4)	1.000 ± 0.056 (IV)
**MIEC-Coated SS**	Mechanical	No	24.06 ± 0.23	63.76 ± 2.59	124.59 ± 2.39	291.44 ± 17.91 (2)	1.430 ± 0.092 (I)
**Ag-AgCl**	Adhesive	Yes	54.88 ± 6.76	115.00 ± 9.43	42.80 ± 2.23	191.77 ± 14.04 (1)	1.220 ± 0.052 (II)
**SS Control**	Mechanical	No	23.17 ± 0.12	65.01 ± 1.13	121.00 ± 4.92	298.02 ± 6.12 (3)	1.000 ± 0.031 (III)

^(a)^ Ranking was established based on the average value of each electrode material concerning the EMG frequency of 100 Hz. Z’: 1–6, lowest to highest Z’; normalized SNR I–VI, highest to lowest normalized SNR. * Values were taken at 103.86 Hz but rounded to 100 Hz in the table.

**Table 3 sensors-24-03136-t003:** Mechanical properties of the MIEC sheet.

Average Modulus (MPa)	Average UTS (MPa)	Average Strain at Break (%)
165.50 ± 21.70	0.85 ± 0.01	567.51 ± 8.45

## Data Availability

The datasets used and/or analyzed by this paper can be obtained from the corresponding author upon reasonable request.

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
