# Peer review of "Benchtop Performance of Novel Mixed Ionic–Electronic Conductive Electrode Form Factors for Biopotential Recordings"

_sensors, 2024, doi:10.3390/s24103136_

Round 1

Reviewer 1 Report

Comments and Suggestions for Authors

The article “Benchtop performance of novel mixed ionic-electronic conductive electrode form factors for biopotential recordings” was very interesting to read.

The researchers have made several forms of their MIEC electrodes to measure physiological properties such as EMG (investigated in this study via simulation), EEG, and EKG.

Authors provide a clear and thorough introduction which contains relevant information on current methods and standards for measuring these properties through a person’s skin.

They very carefully describe the limitations and advantages of both wet and dry electrodes and their decisions on what improvements they sought in fabricating a new interface material for this type of sensing. Their objectives were clear.

The first was to compare the signal and signal to noise ratio of their MIEC electrodes to the standard wet and standard dry electrode. They had a good set of controls and parameters to keep variables at a minimum. Of their four MIEC electrodes (synthesis described succinctly and thoroughly), the “foam” electrode had the most issues and therefore opportunity for improvement, as they state in their conclusion section. The others performed similarly or better than the current standard electrodes.

The graphics are very clear and present the information in a clean and simple manner.

Authors provide in text and through their graphics a clear explanation of their setup and measurements. They also include good references to original references.

The tables summarize well the data and results of measurements, and the writing discusses them at an appropriate length.

Lines 307-310. Were these lines meant to be deleted?

Line 333. Remove “at” from sentence.

For each major concern, after analysis, the authors offer a plan for future work to improve electrodes function and individual applications.

Sentences in Lines 379-388 seem redundant. Preferably one over the other or a more precise way of expressing most important point.

Section 4.3 provides examples of the wider range of applications for these MIEC sensors. Very exciting and medically important .

They were sure to list the differences between electrodes of same type and different days for same readings. The list included common variabilities in all labs such as humidity, and new batch preparation, as well as the human subject skin and temperature.

Very well written. Electrode fabrication, fouling, and comfort to patient/client is a very difficult field. Improvements will have a big impact on the field itself.

Comments on the Quality of English Language

Fine. 2 minor issues. Read above review for exact location in text.

Reviewer 2 Report

Comments and Suggestions for Authors

In this manuscript, the biopotential recording properties of a novel mixed ionic-electronic conductive (MIEC) material was evaluated and improved. This article seems interesting, but the amount of data is insufficient to fully illustrate the advantages of MIEC. The article has not met the threshold for publication on SENSORS for the time being.

1. In the Introduction, the author has analyzed the advantages and challenges of dry electrode and wet electrode for biopotential recording. However, many scholars have studied biopotential recording. Can the author summarize their research? Such modifications will give the reader a quick overview of developments in the field.

2. In the Materials and Methods, the preparation process of MIEC masterbatch is described in detail, and whether it is characterized by relevant materials, such as field emission scanning electron microscopy, etc. The characterization of microstructure can make it easier for readers to understand the properties of electrode materials.

3. It is believed that different MIEC masterbatch production processes have different detection effects. Has the author carried out relevant optimization experiments? If yes, please add, if not, please explain why.

4. The dry MIEC electrode prepared by the authors, whose working scene is on the skin surface of organisms, seems to be more interesting if its mechanical properties are characterized.

5. In the Materials and Methods, The author describes in detail the detection methods of MIEC dry electrodes, such as the placement distance of electrodes, etc., then whether these parameter Settings have an impact on the detection performance and whether there are relevant parameter optimization Settings. If so, please add.

6. The author used artificial simulated skin to detect the biological potential, but in daily life, the skin will be polluted by various pollutants, if the anti-interference research is carried out, then the detection performance of the MIEC dry electrode will be more convincing.

7. The authors should conduct a comparative study with other published biopotential records in terms of analytical performance.

8. The stability of MIEC dry electrodes is worth discussing whether they can be tested all weather.

Reviewer 3 Report

Comments and Suggestions for Authors

It evaluated the bioelectrical recording performance of a novel mixed ion electron conductive (MIEC) material, produced four MIEC electrode shape factors, and compared their signal recording performance with two commonly used control electrodes (Ag AgCl, stainless steel) for bioelectrical recording. Their adaptability and flexibility, discrete contours, reduced weight, and ease of integration into textiles and wearable devices can improve the wearability and comfort of the electrodes.

Comments on the Quality of English Language

none things.

Round 2

Reviewer 2 Report

Comments and Suggestions for Authors

Since my concerns are well addressed, I recommend its publication in present form.